# Glutathione Peroxidase *gpx1* to *gpx8* Genes Expression in Experimental Brain Tumors Reveals Gender-Dependent Patterns

**DOI:** 10.3390/genes14091674

**Published:** 2023-08-24

**Authors:** Cristina Cueto-Ureña, María Jesús Ramírez-Expósito, María Dolores Mayas, María Pilar Carrera-González, Alicia Godoy-Hurtado, José Manuel Martínez-Martos

**Affiliations:** 1Experimental and Clinical Physiopathology Research Group CTS-1039, Department of Health Sciences, School of Experimental and Health Sciences, University of Jaén, 23071 Jaén, Spain; ccueto@ujaen.es (C.C.-U.); mramirez@ujaen.es (M.J.R.-E.); mdmayas@ujaen.es (M.D.M.); pcarrera@ujaen.es (M.P.C.-G.); 2Department of Neurosurgery, Jaén Neurotrauma Hospital, 23009 Jaén, Spain; aliciagodoyhurtado@gmail.com

**Keywords:** brain tumors, glutathione peroxidases, gender, Wistar, oxidative stress, free radicals

## Abstract

Extensive research efforts in the field of brain tumor studies have led to the reclassification of tumors by the World Health Organization (WHO) and the identification of various molecular subtypes, aimed at enhancing diagnosis and treatment strategies. However, the quest for biomarkers that can provide a deeper understanding of tumor development mechanisms, particularly in the case of gliomas, remains imperative due to their persistently incurable nature. Oxidative stress has been widely recognized as a key mechanism contributing to the formation and progression of malignant tumors, with imbalances in antioxidant defense systems being one of the underlying causes for the excess production of reactive oxygen species (ROS) implicated in tumor initiation. In this study, we investigated the gene expression patterns of the eight known isoforms of glutathione peroxidase (GPx) in brain tissue obtained from male and female control rats, as well as rats with transplacental ethyl nitrosourea (ENU)-induced brain tumors. Employing the delta-delta Ct method for RT-PCR, we observed minimal expression levels of *gpx2*, *gpx5*, *gpx6*, and *gpx7* in the brain tissue from the healthy control animals, while *gpx3* and *gpx8* exhibited moderate expression levels. Notably, *gpx1* and *gpx4* displayed the highest expression levels. Gender differences were not observed in the expression profiles of these isoforms in the control animals. Conversely, the tumor tissue exhibited elevated relative expression levels in all isoforms, except for *gpx4*, which remained unchanged, and *gpx5*, which exhibited alterations solely in female animals. Moreover, except for *gpx1*, which displayed no gender differences, the relative expression values of *gpx2*, *gpx3*, *gpx6*, *gpx7*, and *gpx8* were significantly higher in the male animals compared to their female counterparts. Hence, the analysis of glutathione peroxidase isoforms may serve as a valuable approach for discerning the behavior of brain tumors in clinical settings.

## 1. Introduction

Gliomas are the most common type of brain tumors in adults. Those of the higher grade, ranging from CNS WHO grade 2 and 3 oligodendroglioma and astrocytoma to CNS WHO grade 4 astrocytoma and glioblastoma, among others, are the most aggressive and deadly [1]. These tumors have a high degree of genetic and epigenetic heterogeneity, which means that they have different alterations in their DNA and the way that DNA is regulated [2,3,4]. Patients with glioma have a very low chance of survival, with an average survival time of less than 16 months after diagnosis, indicating that the disease is very difficult to treat [5,6]. Many genes have been identified that are involved in the formation and growth of glioma, and these genes can be used as markers to distinguish different types of gliomas and to predict their outcome [5,6]. However, the molecular mechanisms that underlie the development and progression of glioma are still not fully understood, and there is a great need for reliable biomarkers that can help clinicians to make better decisions about diagnosis, treatment, and follow up [7].

One of the key factors that contributes to the development and maintenance of malignant tumors is oxidative stress [8,9]. Oxidative stress is a condition where there is an imbalance between the production and elimination of reactive oxygen species (ROS). ROS are molecules that contain oxygen and are highly reactive, meaning that they can interact with and damage other molecules in the cell. ROS are normally produced by cells as part of their normal metabolism, but they can also increase due to external factors such as radiation or chemicals, or internal factors such as inflammation or hypoxia [10]. Cells have antioxidant systems that can neutralize ROS and prevent oxidative damage. However, cancer cells often have higher levels of ROS production and lower levels of antioxidant defense, which results in oxidative stress. Studies have shown that cancer cells produce a significant amount of ROS and impair the antioxidant system as a result of hypoxia and metabolic changes. Oxidative stress can activate oncogenes, which are genes that can cause normal cells to become cancerous by stimulating cell proliferation, transformation, and metastasis [9,11]. Glutathione (GSH) is the most abundant antioxidant molecule in living organisms, and it has a vital role in removing and detoxifying carcinogens, which are substances that can cause cancer [12]. Targeting GSH metabolism has been suggested as a potential therapeutic strategy to make cancer cells more vulnerable to conventional treatments such as chemotherapy or radiotherapy [10]. Glutathione peroxidase (GPx) includes a family of multiple isozymes that catalyze the reduction of organic hydroperoxides by using reduced GSH as an electron donor [13]. The GPX family has eight isoforms in mammals codified by *gpx1* to *gpx8* genes [13]. In the present work, the gene expression of *gpx1* to *gpx8* in brain tissue from animals with brain tumors induced by the transplacental administration of N-ethyl-N-nitrosourea (ENU) is analyzed in both male and female animals and compared with equivalent brain tissue from healthy animals. The different expression patterns and sex differences found here allow us to propose these genes as biomarkers for the diagnosis and monitoring of this type of tumor.

## 2. Materials and Methods

### 2.1. Animals and Treatments

Ten female and five male Wistar rats, obtained from Harlan laboratories (Spain), were used in this study. The animals were kept in a controlled setting with a temperature of 25 °C and a cycle of 12 h of light and 12 h of darkness. The rats had unrestricted access to food and liquids. The use and care of the animals in the experimental procedures was conducted in compliance with the European Community Council Directive (2010/63/EU). Day 1 of gestation was measured from the time sperm was detected in vaginal smears after male and female Wistar rats were confined together over night. On day 18 of pregnancy, one group of pregnant rats received an intravenous injection of ENU (75 mg/kg body weight), while another group of pregnant rats received an injection of the vehicle (saline solution). The natural delivery and weaning of the offspring occurred at 22 days. The animals at this time were housed and kept separate, being checked weekly for any signs of neurological or physical issues. The surviving rats were killed at the age of 30 weeks.

### 2.2. Magnetic Resonance Imaging (MRI)

The MRI procedures as well as the histology and the parameters of carcinogenesis were previously described [14,15].

### 2.3. Tissue Collection

The rats were anesthetized with equithesin (2 mL/kg body weight) by an intraperitoneal injection and then shaved and sterilized with 10% povidone-iodine. Samples of the tumors were quickly removed and frozen at –80 °C until use. Samples from equivalent brain regions were obtained from the control animals according to Paxinos and Watson [16]. RNA was extracted from all the samples by using a Trizol-based RNA isolation protocol, followed by cDNA synthesis and amplification with commercially available kits.

### 2.4. RT-PCR

For first-strand cDNA synthesis, the total RNA was reverse transcribed by using random hexamers (Roche Diagnostic, Barcelona, Spain) as primers and Transcriptor Reverse Transcriptase (Roche Diagnostic, Barcelona, Spain). Gene expression was assessed by RT-PCR (BioRad, Madrid, Spain) with SYBR Green detection. For quantification, we used the delta-delta Ct method with two reference genes (β-actin and GAPDH). All the samples were quantified in duplicate. The primers’ design is shown in Table 1.

## 3. Results

Figure 1 shows the *gpx1* to *gpx8* expression level in the brain tissue obtained from the healthy male and female control animals. *gpx2*, *gpx5*, *gpx6*, and *gpx7* exhibited extremely low expression levels, while *gpx3* and *gpx8* displayed moderate expression levels. The highest expression levels were observed for *gpx1* and *gpx4*. No significant differences in the expression levels of these genes were observed between the male and female healthy control animals.

In contrast, as shown in Figure 2, the tumor tissue displayed significantly (*p* < 0.001 in all cases) increased relative expression levels for all glutathione peroxidase genes, except for *gpx4* (Figure 2D), which remained unchanged, and *gpx5* (Figure 2E), which exhibited changes only in the female animals. Notably, except for *gpx1* (Figure 2A), which showed no significant gender differences, the relative expression values of *gpx2* (Figure 2B), *gpx3* (Figure 2C), *gpx6* (Figure 2F), *gpx7* (Figure 2G), and *gpx8* (Figure 2H) were significantly higher (*p* < 0.001) in the male animals when compared to the female animals.

## 4. Discussion

The equilibrium between the production and elimination of ROS is regulated by several enzyme antioxidant defenses, such as superoxide dismutase (SOD), catalase (CAT), and GPx. Their activities avoid or reduce the damage caused by free radicals to macromolecules such as lipids, proteins, and nucleic acids [9,14]. Particularly, GPx is responsible for reducing H_2_O_2_ and other organic hydroperoxides into water and alcohols, respectively [13]. During chronic or excessive oxidative stress, changes in antioxidant enzyme activities may occur, which may or may not be sufficient to avoid oxidative damage and may play a key role in the pathogenesis of several diseases including cancer. In this study, we analyze the expression of *gpx1* to *gpx8* genes for the eight known GPx isoforms in brain tissue from healthy control animals and tumor tissue from animals after transplacental ENU administration, also considering gender differences.

GPx-1 is an important isoenzyme for the protection of the organism against systemic oxidative stress [17]. Several in vivo studies show that a GPx-1 deficiency causes cell damage, apoptosis, and even cell death. In mice, studies of traumatic brain injury showed that a GPx-1 deficiency potentiates free radical damage in hippocampal neurons, impairing memory [18]. The same occurs in cochlear spiral ganglion neurons. Here, a lack of GPx-1 causes oxidative damage leading to the death of these neurons and hearing loss [19]. The lack of GPx-1 exacerbated the damage caused by oxidative stress in multiple organs and tissues; therefore, it is expected that brain-tumor-induced damage promotes increased levels of *gpx1* in our experimental model. Indeed, other studies demonstrate that the increased expression of *gpx1* has protective effects in various organs and tissues. In a model of brain damage, GPx-1 overexpression facilitates spatial learning in young animals because oxidative damage is reduced from early stages [20]. Regarding cancer, it has been extensively described that GPx-1 protein levels and *gpx1* mRNA expressions are upregulated in tumor (carcinoma) tissues to protect against DNA damage due to their association with an increased risk of cancer. Significant increases in enzyme activity, GPx-1 protein levels, and *gpx1* mRNA expression in male and female rats with ENU-induced tumors have been previously described by us [14]. Several authors confirmed that GPx-1 plays an important role in protecting neural cells from oxidative stress [21,22,23], reducing hydrogen peroxide to water to limit its harmful effects. Other authors [24,25] found that a GPx-1 deficiency exacerbates neurotoxicity in cortical neurons and increases infarct size in response to ischemia/reperfusion injury, respectively. Overall, GPx-1 is the most widely studied enzyme, and its important antioxidant activity seems to play a crucial role in protecting the brain from oxidative stress and neurodegeneration, in agreement with our results.

On the other hand, GPx-2 is both a selenium-containing and GSH-dependent enzyme that is enriched in the gastrointestinal tract [11]. GPx-2 could also reduce hydroperoxides and promote defense against oxidative stress [12,13]. The role of GPx-2 in cancer has been described in the last few years. Thus, it has been shown that increased GPx-2 is related to the promotion, growth, and metastasis of colorectal, hepatocellular, and bladder cancers [12,14,15]. In fact, the role of GPx-2 in forming a multigene prognosis model for glioblastoma multiforme has been suggested, where a significant association of higher *gpx2* expression with poorer prognosis was detected. Our results also showed increased *gpx2* expression in the tumor tissues of males and females, with important gender differences. Other authors also found that a prognostic analysis depends on gender. This supports the idea that GPx-2 effects are gender specific. Sex steroids, which are bioactive molecules with hormonal properties that regulate various physiological and developmental functions in the body, as well as sex chromosomes, structural components of the genome that carry genetic information relevant to sex determination and the regulation of gene expression linked to sexual development and reproduction, can exert highly influential intrinsic control over cancer-initiating cell populations, the microenvironment surrounding the tumor, and the determinants who facilitate the development of cancer [22]. In fact, several studies demonstrated that there is a sophisticated, bidirectional interaction between sex steroids, sex chromosomes, and cellular and molecular processes associated with carcinogenesis, which also involved oxidative stress and cell metabolism [23,24]. Altogether, this supports the idea that sex steroids may be involved in the GPx-2 tumor-related functions. An important aspect related to GPx-2 is that it has been identified as a target of the Wnt/β-catenin pathway [26]. The Wnt pathway regulates the gene expressions that stem cells need to proliferate in certain locations to maintain their homeostasis. Thus, during an inflammatory process, inflammatory cells release oxidative substances that function as signals to stimulate ROS production. In the Wnt pathway, the main step is the binding of Wnt to its Frizzled receptor so that the nuclear redox protein (NRX), which blocks Disheveled (DVL), plays a negative regulatory role [27]. When NRX is oxidized, the release of DVL avoids the formation of components of the β-catenin degradation complex, stabilizing it and activating the *gpx2* gene [28]. The increase in GPx-2 favors the elimination of H_2_O_2_, the inhibition of apoptosis, and the decrease in oxidative stress, generating a feedback loop that regulates the activity of the Wnt pathway itself. GPx-2 is, therefore, a protective enzyme against oxidative stress that also prevents and reduces the inflammatory response so that it can stop tumorigenesis but favor the growth of existing tumors. It has therefore been proposed that *gpx2* expression levels depend on the phase in which it is found. Further research is necessary to understand how hormones could influence *gpx2* regulation and functions.

GPx-3 is an enzyme present at the extracellular level on the external location of plasma and mucosal membranes and in the cytoplasm and plasma membranes of various tissues, including the brain, protecting cells from oxidative damage [29]. Its best-known function with respect to cancer is to inhibit the expression of HIF1-α through the regulation of ROS to inhibit melanoma cell growth [30]. In addition to an increase in inflammation levels, the number of tumor cells also increased during the development of colitis-related cancers in *gpx3* knockouts [31]. This may be related to the GPx-3 inhibition of oxidative stress and apoptosis levels. Other authors found that GPx-3 inhibits oxidative-stress-related pathways (the NFκB pathway) and apoptosis-related pathways (the caspase pathway). In the same way, GPx-3 also seems to be regulated by estrogens [32], but to our knowledge, no information is available about its role in brain tumors, although several important functions of GPx-3 have been addressed in several brain pathologies. In our study, *gpx3* was highly expressed in brain tumor tissue and also showed gender differences in the same way that *gpx2* did.

Glutathione peroxidase 4 (GPx-4) has also been involved in many diseases. It reduces phospholipid hydroxides, avoids lipoxygenase overactivation, and prevents lipid peroxidation [33,34,35]. The genetic knockdown of GPx-4 increased the cytotoxicity of oxytosis induced by glutamate in the retina in terms of lipid peroxidation and apoptosis [36]. Also, GPx-4 protects mitochondria from oxidative damage in gut epithelial cells [37]. In Parkinson’s disease, altered GPx-4 expression and distribution has been associated with pathological changes in the brains of patients, supporting the idea that high GPx-4 levels could protect neurons against oxidative damage [38]. Recently, it has been described that ferroptosis, a form of regulated necrosis [39], is mediated by GPx-4 [40,41]. It is characterized by metabolic alterations that promote excessive ROS generation through an iron-dependent pathway [42,43]. Ferroptosis seems to be due to the inactivation of GPx-4 through GSH depletion, which promotes the accumulation of ROS from lipid peroxidation. Some morphological characteristics of ferroptosis are cell volume shrinkage, a lack of rupture and blebbing of the plasma membrane, and small mitochondria with condensed membrane densities lacking the traditional apoptotic and necrotic phenotypes [44,45,46]. However, in our animal model, we found no differences in *gpx4* gene expression in either the male or female animals. Despite being the gene of the glutathione peroxidase family that is mostly expressed in our brain tissue, it does not seem that the appearance of brain tumors modifies its function.

To our knowledge, GPx-5 has not been previously described in brain tissue. In fact, in healthy animals, its expression is extremely low. This enzyme has mainly been described in the male reproductive tract, which suggests its dependence on sex hormones [13]. In fact, we have not found changes in *gpx5* expression in males with brain tumors, whereas in females with brain tumors, an important increase in the expression of this enzyme appears. Further research is necessary to clearly evaluate the functional role of brain GPx-5 in male and female rats. However, GPx-5 plays an important role in the maintenance of the tissue microenvironment, cellular protection against oxidative stress, and maintenance of the DNA structure integrity [47], which needs to be studied at the brain tissue level as a consequence of tumor development.

GPx-6 is a protein closely homologous to GPx-3. Recent studies have shown that the increased expression of GPx-6 protein can decrease oxidative stress [48]. Also, the study of the neuroprotective effect of mesenchymal stem cells on the regulation of oxidative stress in retinal nerves showed that GPx-6 expression was upregulated [49]. Both studies suggest its possible antioxidant function. However, the mechanism of action of GPx-6 is not known. Moreover, it has been described that detoxification enzymes such as GPx-6 significantly increased in the early stage of neurodegenerative diseases such as Parkinson’s and Huntington’s diseases as a compensatory mechanism for the increment of various toxic factors after the onset of the disease [50,51]. However, its role in brain tumors is unknown, beyond its role as a detoxifying enzyme. Here we have found a pattern of expression like other GPxs.

Glutathione peroxidase 7 (GPx-7) is an enzyme localized in the endoplasmic reticulum. It has also been called nonselenocysteine phospholipid hydroperoxide glutathione peroxidase and uses H_2_O_2_ as an oxidant [52,53,54,55]. Physiologically, GPx-7 is an intracellular sensor that detects redox levels and transmits ROS signals to its interacting proteins by disulfide bond shuttling. Several biological process are involved, such as oxidative protein folding [56,57,58], the release of nontargeting short interfering RNAs (siRNAs)-associated stress [59,60,61], and protection against oxidative stress [56,59,62,63,64]. GPx-7 has also been related to metabolic diseases [62,65,66], neurodegeneration [67,68,69], viral infection [70], and cardiovascular diseases [61]. Furthermore, the malfunction of GPx-7 contributes to tumorigenesis and the progression of diverse types of carcinomas in humans. Examples are esophageal adenocarcinoma [71], gastric cancer [72], hepatocellular carcinoma [73], acute myeloid leukemia [74], and breast cancer [75]. The expression patterns of GPx-7 have been described from a pancancer perspective, using microarray data through the analysis tool of the Oncomine database [52]. The authors describe that GPx-7 was overexpressed in brain, breast, esophageal, gastric, and liver cancers as well as leukemia, melanoma, myeloma, and sarcoma and underexpressed in lymphomas. They conclude that the dysregulation of GPx-7 was a common phenomenon across several tumors. However, the role of GPx-7 in the development of brain tumors is not well known. It has been described that GPx-7 is higher in gliomas, especially in astrocytic, oligodendroglial, and mixed gliomas, supporting the data described here and settling the idea that *gpx7* overexpression may be involved in glioma tumorigenesis. This process seems to be differentially regulated by molecular mechanisms during the cell cycle. It is possible that numerous factors of the tumor microenvironment, in response to exposure to oxidative stress, modulate this expression pattern. In fact, GPx-7 expression was higher in glioblastoma than in low-grade gliomas (oligoastrocytoma, oligodendroglioma, and astrocytoma), while the expression of GPx-7 was higher as the pathological grade increased. Furthermore, GPx-7 is overexpressed in aggressive gliomas, which also suggests that GPx-7 may be involved in the malignant progression of gliomas. These authors conclude that the overall GPx-7 expression varied significantly as a function of histopathological grade, is highly associated with glioma progression, and correlates with an unfavorable prognosis in adult low-grade glioma tumors. However, no gender differences have been described for *gpx7* expression that could explain the results found in our study, as has occurred for other members of the GPx family.

Finally, GPx-8 is the last discovered member of the GPx family. It is a transmembrane protein located, as GPx-7, in the endoplasmic reticulum. In fact, GPx-8 is similar to GPx-7 in that both have low glutathione peroxidase activity while having a function related to promoting the oxidative folding of endoplasmic reticulum proteins and reducing oxidative stress [76]. It contains an N-terminal signal peptide and a C-terminal endoplasmic reticulum membrane localization signal and functions in regulating calcium homeostasis [77,78]. Importantly, the overexpression of GPx-8 reduces histamine-induced Ca^2+^ storage and release in the endoplasmic reticulum [77]; thus, GPx-8 overexpression results in decreased Ca^2+^ levels in the endoplasmic reticulum while its silencing causes Ca^2+^ release from the endoplasmic reticulum to the mitochondria and cytoplasm [77]. In addition, GPx-8 is linked to various sorts of human malignant tumors [79,80,81,82]. The overall result revealed that GPx-8 was highly expressed in gliomas, as we have also described here. In the same way, the existence of gender differences in *gpx8* expression needs further research to elucidate its functional role.

The findings of our study indicate that the analysis of glutathione peroxidase isoforms, including those with extremely low expression in healthy tissue, may hold promise to discern the behavior of brain tumors in clinical settings. The observed alterations in gene expression profiles, particularly the significant increase in male animals but also the existence of gender differences, provide insights into the potential roles of these isoforms in tumorigenesis and suggests gender-dependent differences in tumor development mechanisms.

## 5. Limitations and Future Directions

This study has two important limitations. Firstly, the number of animals used in our experiments. It should be considered that these studies start with a large number of animals, but due to the high mortality rate of brain tumors induced by ENU, the final number of animals available at the end of the experiment is relatively low. A balance must be found between the appropriate use of laboratory animals and the significance of the results, always considering the 3R principle and following the guidelines of the regulations of the bioethics committees. Nevertheless, from a statistical power point of view, the significances are large enough to conclude the existence of differences between groups.

Secondly, due to the fact that epigenetic modifications can be of great relevance to understand the behavior of the products of the genes that have been analyzed, the analysis of protein expression is an important step for the elucidation of the functions of the different GPx in brain tumors, as well as the study of the post-transcriptional modifications that may occur. In fact, in a previous study in our laboratory, we had already shown that *gpx1* gene expression levels were increased in male and female animals with brain tumors, correlating these values with protein expression levels and enzyme activity in the tumor tissue, while no significant change was found in enzyme activity at the plasma level, which could represent a pool of GPx isoforms of different origins. However, we strongly believe that the results shown in the present work are of enormous importance precisely because (1) they highlight the different patterns of gene expression for the different types of GPx, implying the alteration of the mechanisms of regulation of antioxidant enzymatic defenses in brain tumors independently of the subsequent modifications that may occur, and (2) they show that some of these alterations are gender dependent; new study possibilities are opened to determine the importance that circulating sex steroids may have in the regulation of antioxidant mechanisms in these tumor processes, as well as many other additional studies that can be carried out.

Finally, it should also be pointed out that brain tumors are probably one of the least known tumors and have the least available treatments and therefore are the most in need of study. Any indication or any new contribution that can be made to deepen our knowledge or at least lay the foundation for further studies is vital, as the goal is to achieve a better understanding of tumors as aggressive as brain tumors; because we have so little information on their pathophysiological processes, their treatment is currently very limited.

## 6. Conclusions

Our study reveals differential gene expression patterns of glutathione peroxidase genes in brain tissues from control animals and animals with brain tumors induced by the transplacental administration of ENU. The significant upregulation of multiple isoforms in the tumor tissue, particularly in the male animals, suggests their potential as discriminative markers for brain tumor behavior. This research contributes to the understanding of oxidative stress mechanisms and highlights the importance of gender-dependent differences in brain tumor development. Further investigation and validation of these findings in human brain tumor samples could determine the clinical utility of glutathione peroxidase isoforms as biomarkers for brain tumor diagnosis and prognosis.

## Figures and Tables

**Figure 1 genes-14-01674-f001:**
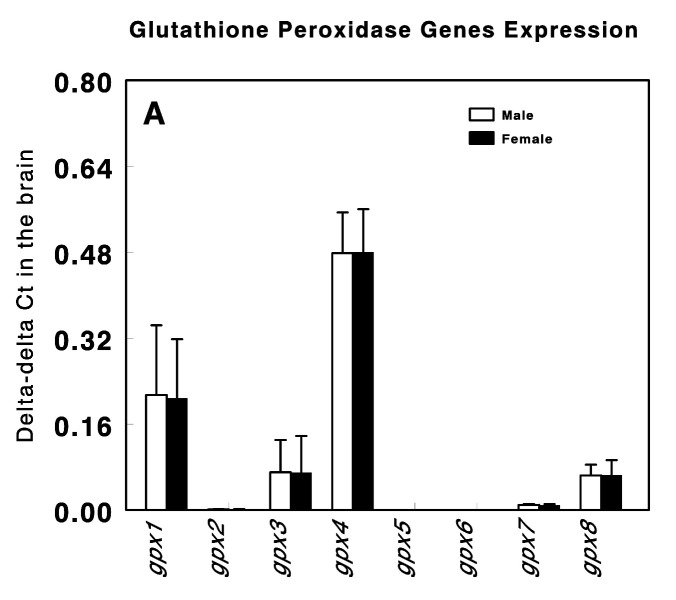
Double-delta Ct values for *gpx1* to *gpx8* genes expression in brain tissue of healthy control male and female Wistar rats. Values are expressed as Mean ± SEM.

**Figure 2 genes-14-01674-f002:**
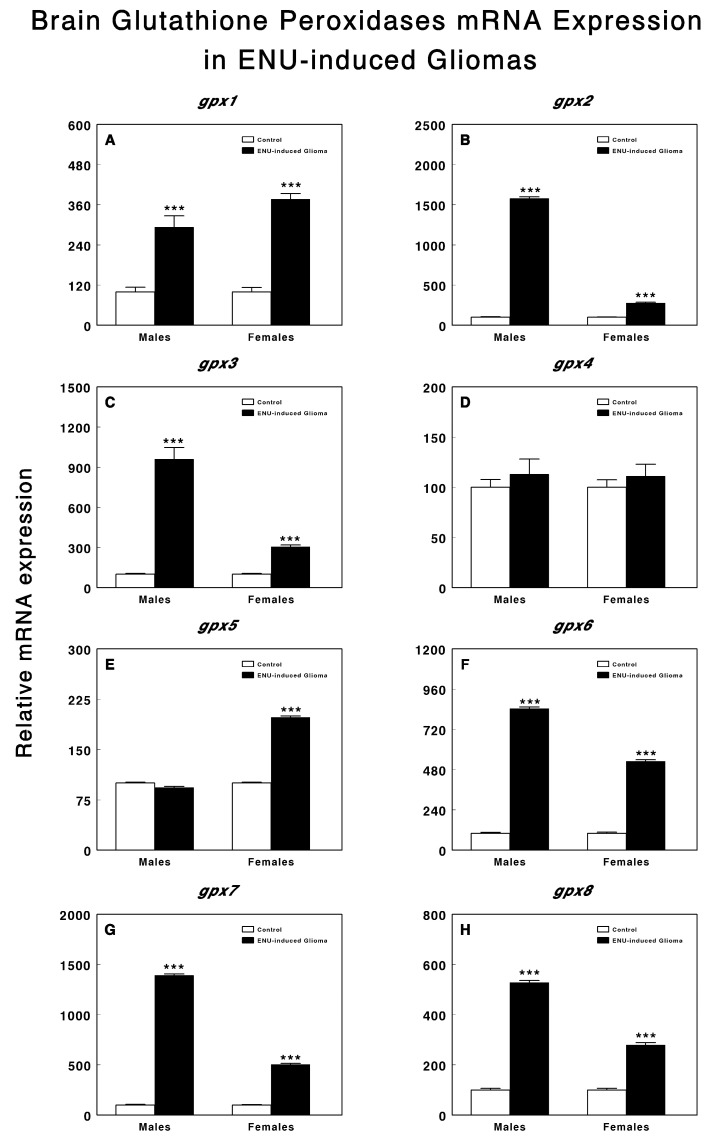
*gpx1* (**A**), *gpx2* (**B**), *gpx3* (**C**), *gpx4* (**D**), *gpx5* (**E**), *gpx6* (**F**), *gpx7* (**G**), and *gpx8* (**H**) mRNA relative expression levels in brain tissue of control male and female rats and in tumor tissue of male and female rats with ENU-induced gliomas. Data were normalized to ß-actin and GAPDH levels. Values are expressed as relative mRNA expression (Mean ± SEM; *** *p* < 0.001).

**Table 1 genes-14-01674-t001:** Primers used for quantitative real-time PCR.

Wistar Rat Gene	Forward Primer 5′→3′	Reverse Primer 5′→3′
*gpx1*	AGTTCGGACATCAGGAGAATGGCA	TCACCATTCACCTCGCACTTCTCA
*gpx2*	GACACGAGGAAACCGAAGCA	GGCCCTTCACAACGTCT
*gpx3*	CAGAACTCCTGGGCTCACCT	TCCATCTTGACGTTGCTGAC
*gpx4*	CCGGCTACAATGTCAGGTTT	ACGCAGCCGTTCTTATCAAT
*gpx5*	CTTGGATCCGAATTCATGACCCCCAGGCT	CTGCAGGATCCTAAGCTTATATGGTTTTGA
*gpx6*	CAGAAGTTGTGGGGTTCCTGT	TGCCAGTCACCCCTTTGTTG
*gpx7*	CCTGCCTTCAAATACCTAACCC	TGTAATACGGGGCTTGATCTCC
*gpx8*	CTACGGAGTAACTTTCCCCATCTTCCACAAG	CTGCTATGTCAGGCCTGATGACTTCAATGG
ß-Actin	CTCTCTTCCAGCCTTCCTTC	GGTCTTTACGGATGTCAACG
GAPDH	GCACCGTCAAGGCTGAGAAC	ATGGTGGTGAAGACGCCAGT

## Data Availability

The data presented in this study are available on request from the corresponding author.

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
