# Peer review of "Glutathione Peroxidase gpx1 to gpx8 Genes Expression in Experimental Brain Tumors Reveals Gender-Dependent Patterns"

_genes, 2023, doi:10.3390/genes14091674_

Round 1

Reviewer 1 Report

The study investigated the oxidative stress genes change patterns in both male and female animals with/without ENU induced brain tumor which is a strength of the study. However, the study does not sufficiently explore the underlying mechanism behind the reported findings. Understanding the mechanism is critical to establish the scientific basis and significance of the results. I recommend conducting further investigations and providing a more comprehensive analysis of the mechanism to strengthen the research.

The work presented in the manuscript appears to be relatively preliminary and insufficient and may require further in-depth experimentation and analysis. For publication in the journal, I expect a higher level of scientific rigor and a more substantial contribution to the field.

The format of gene names should be corrected.

Author Response

Reviewer #1.

We would like to sincerely thank the reviewer for his/her comments.

As I have told the Editor beforehand, epigenetic modifications can be of great relevance to understand the behavior of the products of the genes that have been analyzed, so that the analysis of protein expression is an important step for the elucidation of the functions of the different GPx in the brain tumor, as well as the study of post-transcriptional modifications that may occur.  In fact, in a previous study in our laboratory we had already shown that gpx1 gene expression levels were increased in male and female animals with brain tumors, correlating these values with protein expression levels and enzyme activity in the tumor tissue, while no significant change was found in enzyme activity at the plasma level, which could represent a pool of GPx isoforms of different origins. Indeed, this same study is possible with the rest of GPx. Unfortunately, our laboratory does not currently have the funding to carry out these additional studies. However, we strongly believe that the results shown in the present work are of enormous importance precisely because 1) they highlight the different patterns of gene expression for the different gpx implying the alteration of the mechanisms of regulation of antioxidant enzymatic defenses in brain tumors independently of the subsequent modifications that may occur; and 2) show that some of these alterations are gender-dependent; even new study possibilities are opened to determine the importance that circulating sex steroids may have in the regulation of antioxidant mechanisms in these tumor processes, as well as many other additional studies that can be carried out. But this work is pioneering to lay the foundations and be the reference for future studies by other authors. Finally, it should also be pointed out that brain tumors are probably one of the least known tumors and with the least available treatments and therefore most in need of study. Any indication or any new contribution that can be made to deepen our knowledge, or at least to lay the foundations for further studies, is, in our opinion, of sufficient importance. We are aware that many questions remain to be answered. We will be able to in the future, but many others will have to be investigated by other research groups that want to pick up and expand on the results that we show in this article. In the end, the important thing is to achieve a better understanding of tumors as aggressive as brain tumors, for which we have so little information on their pathophysiological processes that their treatment is currently very limited.

Reviewer 2 Report

This is a well-conducted study about changes of glutathione peroxidase gpx1 to gpx8 gene expression in experimental brain tumors.

The following points have to be raised:

- Not all gliomas are aggressive and deadly as described in lines 34 and 35. The authors should differentiate between astrocytomas WHO grade 2, 3 and 4, GBM WHO grade 4, oligodendrogliomas WHO grade 2 and 3.

- Please provide more information about how many animals developed ENU induced brain tumors.

- In which location were the tumors found, what was their size and their histological appearance. Provide pictures of the histology.

- Provide an MRI-scan

- Please explain why you did not use glioma tissue obtained intraoperatively while one co-author is a neurosurgeon. This would provide data which reflect the real situation in contrast to the experimental group.

minor corrections

Author Response

Reviewer #2.

              We would like to sincerely thank the reviewer for the helpful suggestions offered as means for improving the paper.

  • According to reviewer #2 suggestion, a reference to brain tumors WHO classification has been included in the revised version of the manuscript.
  • Related to the question of why we did not use glioma tissue obtained operatively, we must say that it is not possible due to methodological limitations, mainly sample size. The rat brain needs to be fully extracted and carefully dissected.
  • Please note that the information you are requesting regarding carcinogenesis parameters, localization, histology or MRI, among other data, has been previously published (please see references 14 and 15) and we do not believe it is appropriate to include it again in this article to avoid copyright infringements.

Reviewer 3 Report

Dear Author,

I appreciate your work and the scientific work title "Glutathione peroxidase gpx1 to gpx8 genes expression in experimental brain tumors reveal gender-dependent patterns". The work on the gpx1 to gpx8 gene using a mouse model and gene expression study on the genes in brain tissue is well written with clear aims and objectives, the experimental model is well explained and the results are straightforward and support the research objectives. The gender difference in gene expression pattern Wistar rats, the only question is that with only 15 rats in total is it worth concluding the gender difference?

Thank you 

Author Response

Reviewer #3.

We would like to sincerely thank the reviewer for his/her comments.

Certainly, the number of animals used in our experiments is one of the limitations of our study. As is usually the case with all studies of these characteristics, a balance must be found between the appropriate use of laboratory animals and the significance of the results, always considering the 3R principle and following the guidelines of the regulations of the bioethics committees. It should be considered that these studies start with a large number of animals, but due to the high mortality rate of brain tumors induced by ENU, the final number of animals available at the end of the experiment is relatively low.  Nevertheless, from a statistical power point of view, the significances are large enough to conclude the existence of gender differences.